# Transient Hyperinsulinemic Hypoglycemia Linked to *PAX6* Mutation

**DOI:** 10.3390/medicina57060582

**Published:** 2021-06-07

**Authors:** Jee-Min Kim, Seul-Ki Kim, Shin-Hee Kim, Won-Kyoung Cho, Kyoung-Soon Cho, Min-Ho Jung, Byung-Kyu Suh, Moon-Bae Ahn

**Affiliations:** Department of Pediatrics, College of Medicine, Catholic University of Korea, Seoul 06591, Korea; jeejennykim@gmail.com (J.-M.K.); seulki12633@gmail.com (S.-K.K.); tigger1018@naver.com (S.-H.K.); wendy626@catholic.ac.kr (W.-K.C.); soon926@catholic.ac.kr (K.-S.C.); jmhpe@catholic.ac.kr (M.-H.J.); suhbk@catholic.ac.kr (B.-K.S.)

**Keywords:** paired box-6 gene, aniridia, congenital hyperinsulinism, glucose intolerance, diabetes mellitus

## Abstract

Prolonged hyperinsulinemic hypoglycemia in infancy can result in developmental sequelae. A mutation in the paired box-6 gene (*PAX6*) has been reported to cause disorders in oculogenesis and neurogenesis. A limited number of cases of diabetes mellitus in adults with a *PAX6* mutation suggest that the gene also plays a role in glucose homeostasis. The present case report describes a boy with a *PAX6* mutation, born with anophthalmia, who underwent hypoglycemic seizures starting at 5 months old, and showed a prediabetic condition at 60 months. This patient provides novel evidence that connects *PAX6* to glucose homeostasis and highlights that life-threatening hypoglycemia or early onset glucose intolerance may be encountered. The role of *PAX6* in glucose metabolism and insulin regulation should be further investigated.

## 1. Introduction

Hyperinsulinemic hypoglycemia (HH) is the most common cause of recurrent hypoglycemia during the neonatal and infantile periods, and excessive glucose utilization due to either exogenous or endogenous hyperinsulinism may lead to severe neurological sequelae. Primary HH is triggered by mutations in genes related to the development of the pancreas resulting in the inappropriate secretion of insulin, and approximately 15 genes, including *ABCC8, KCNJ11, GLUD1, GCK, HADH,* and *SLC16A1*, have been identified [1]. Secondary HH can occur from a variety of syndromes, including intrauterine growth retardation, maternal diabetes mellitus (DM), birth asphyxia, and postoperative complications of gastrointestinal surgery [2]. In patients diagnosed with HH, it is important to develop an early intervention to prevent permanent neurological damage from prolonged hypoglycemia.

The role of the paired box-6 gene (*PAX6*) and its encoding proteins has provided insight into various transformative discoveries by their biological and molecular action as master regulatory proteins and their involvement with oculogenesis, pancreatogenesis, and neurogenesis [3]. With *PAX6* mutations, a number of cases associated with aniridia have been documented, while only a few cases associated with impaired glucose tolerance or DM, which are potential consequences of *PAX6* action on the biosynthesis and secretion of insulin, have been reported [4,5]. In animal models, *PAX6* is known to play an essential role in the differentiation and function of β-cells [6].

Proper *PAX6* expression is important for maintaining β-cell identity and glucose homeostasis; therefore, *PAX6* should be one of the candidate genes to evaluate when impaired glucose tolerance or DM is presented in congenital form. Hyperglycemia induced by low insulin secretory capacity is a theoretical rationale; on the other hand, insulin hypersecretion seems challenging to explain. While studies examining the relationship between HH and *PAX6* are rarely reported, we describe, for the first time, a child with multiple hypoglycemic episodes due to hyperinsulinemia possibly triggered by a novel *PAX6* mutation.

## 2. Case Description

A male infant was born at 40 + 5 weeks of gestation, and was the only child born to Korean parents with no family history of consanguinity or past medical history. The patient’s birth weight was 3.3 kg (25th–50th percentile) and head circumference was 34 cm (25th–50th percentile). He was delivered by normal vaginal delivery during spontaneous labor. He was born in good condition with normal Apgar scores and no antenatal concerns, except for bilateral anophthalmia noticed by fetal ultrasound. Both eyeballs were missing; therefore, the patient was referred to an ophthalmologist soon after birth to discuss ocular prosthesis and was subsequently discharged home. No dysmorphic features including skin pigmentations, umbilical hernia, or hepatomegaly were observed. A physical examination revealed a normal prepubertal male genitalia with Tanner stage 1. Abdominal sonography detected no hepatomegaly or other structural abnormality in the kidney or pancreas. He did not have remarkable syndromic features other than anophthalmos and multiple hypoglycemic seizure episodes.

The patient was 5-months old when he first visited the emergency room (ER) due to an afebrile generalized tonic-clonic seizure lasting 2 min at home. The seizure ceased upon ER arrival, and the patient was mentally alert, showing good activity with no specific findings upon physical examination. Serum was immediately collected, and the serum glucose was 25 mg/dL, with increased insulin and c-peptide in the absence of metabolic acidosis, which was suggestive of HH. Serum β-hydroxybutyrate, free fatty acids, lactate, ammonia, cortisol, and thyrotropin levels were normal (Table 1). No urine ketone was present.

2 mL/kg of 10% dextrose fluid was intravenously administered and the blood glucose level shortly normalized to 109 mg/mL, which was considered to be euglycemic. No family history of hypoglycemia, DM, or other conditions associated with glucose intolerance were documented. The elevation of serum glucose and insulin levels was observed (45 mg/dL and 13.1 μU/mL, respectively) after post-glucagon stimulation (0.03 mg/kg). Brain magnetic resonance imaging (MRI) identified no abnormal signal intensities or structural anomalies, except for nearly absent bilateral orbital globes with atrophic optic nerves, while pancreas MRI detected no structural defect (Figure 1). Sanger sequencing of *ABCC8*, *KCNJ11*, *GCK*, and *BMP4* identified no pathogenic variants. Soon after starting oral diazoxide (5 mg/kg/day divided into 3 doses), a blood glucose level within the normoglycemic range (80–120 mg/dL) was achieved. The parents were advised to regularly monitor blood glucose level, promptly react to the patient’s irritability from long-term fasting, and feed him frequently. The parents refused to apply continuous glucose monitoring sensors, and thus mandatory and routine monitoring of pre- and postprandial glucose was recommended. The patient was discharged, without experiencing additional hypoglycemic episodes.

During follow-ups in the outpatient clinic, the patient showed normal growth and reached the neurodevelopmental milestone of his age (he was able to read Braille). Neither hypo- nor hyper-glycemia associated symptoms were observed while self-monitoring of blood glucose checks maintained within a stable range (70–100 mg/dL for pre-prandial and 150–180 mg/dL for postprandial). Follow-up insulin, c-peptide, and glycated hemoglobin (HbA1c) levels (5.5–8.5 µU/mL, 0.7–1.1 ng/mL and 4.6–5.5%, respectively) remained within the euglycemic range. 5 mg/kg/day of diazoxide was sufficient for the patient to stay euglycemic, without the need for escalating the dose or adding octreotide. Until 36 months of age, he did not show signs of any adverse effects to diazoxide. We decided to slowly taper out the dosage of diazoxide to see whether the condition was temporary since there was no genetic evidence associated with permanent hyperinsulinemic hypoglycemia while the patient remained euglycemic. A weekly reduction of 1 mg/kg took place and diazoxide administration was completely stopped at 38 months of age.

The patient revisited the ER for the 2nd hypoglycemic attack after 4 weeks of diazoxide discontinuation. Serum glucose of the critical sample was 38 mg/mL, and other biochemical results again demonstrated potential HH (Table 1). Emergent glucose infusion and re-administration of 5 mg/kg/day diazoxide resulted in recovery from hypoglycemia. Without recurrence of additional hypoglycemic episodes, the patient was discharged and continued on minimum dose diazoxide. Targeted next generation sequencing (NGS) of 20 genes associated with congenital diabetes (*ABCC8, BLK, CEL, EIF2AK3, FOXP3, GATA4, GATA6, GCK, HNF1A, HNF4A, HNF1B, INS, KCNJ11, KLF11, NEUROD1, PAX4, PAX6, PDX1, PTF1A*, and *ZFP5*) was performed, and a heterozygous variant of 188th nucleotide change from cytosine to guanine, resulting in a change of the 63rd amino acid sequence, serine to cysteine at exon 6 of *PAX6* (NM_00280.4) was identified; this was confirmed by Sanger sequencing. According to the ACMG guideline, the alternation was classified as pathogenic since it was located in a mutational hot spot or well-studied functional domain without benign variation, absent/rare in population databases, and novel missense change at an amino acid residue where a different missense change determined to be pathogenic had been seen before [7,8].

Diazoxide therapy was reinitiated at 39 months and was continued for 2 years; further hypoglycemic events were not observed. We discussed a reduction in diazoxide dosage with the parents; however, they were afraid to do so due to previous seizure episodes. At the age of 60 months, the patient had intermittent hyperglycemia (>200 mg/dL), and laboratory findings on fasting state indicated a prediabetic condition (Table 1). Anti-pancreatic antibodies against glutamic acid decarboxylase, islet antigen 2, zinc transporter, and islet cell were not detected. Intermittent hyperglycemia and borderline HbA1c level (6.1%) presenting at the minimum dose of diazoxide led to concern of possible DM, although the mechanism is yet to be explained in mutations presenting as hyperinsulinemic hypoglycemia progressing to DM. Thus, diazoxide was reduced every other day. The patient remained asymptomatic and showed euglycemic trends during dose tapering. After cessation of diazoxide, the 6- and 12-month fasting glucose and HbA1c levels were 86 and 5.6% and 89 and 5.5%, respectively.

## 3. Discussion

*PAX6* spans 23 kb on 11p13 and includes 14 exons and 422 amino acids [8]. Since its discovery in 1991, *PAX6* studies have spanned many decades and have unveiled the biological roles and molecular mechanisms of associated proteins. In a wide spectrum of impaired ocular morphogenesis, aniridia is a casual consequence of heterozygous loss-of-function mutations in *PAX6* [9]. Approximately 80% of congenital aniridia are due to *PAX6* mutations, and more than 300 mutations have been identified to date [10,11]. The role of *PAX6* in glucose metabolism has recently been studied; hypoglycemia or DM may be triggered in patients with *PAX6* mutations, by aberrant glucagon and insulin signaling pathways of pancreatic α- and β-cells [1,12]. Among published *PAX6*-induced congenital aniridia cases, this is the first report demonstrating a transitional pattern from hypoglycemia to hyperglycemia in early childhood.

Congenital HH is a genetic disorder characterized by either diffuse or focal abnormalities of the islets of Langerhans, with an incidence ranging from 1/50,000 to 1/2500 live births [13]. Less than 20 candidate genes associated with congenital HH have been identified [2]. Even though studies have been conducted to reveal the role of *PAX6*, its direct mechanism regulating glucose homeostasis is still unknown. Nevertheless, evidence from both animal and human studies has suggested that genetic alteration of *PAX6* affects islet function, which in turn regulates glucose metabolism [3,14]. According to Hart et al., glucagon production was significantly reduced in response to inactivation of *PAX6* expression [15]. Ablation of *PAX6* in mouse pancreatic endocrine cells resulted in a reduction of cells secreting insulin and glucagon [15]. Mice with heterozygous *PAX6* mutation had a prohormone convertase 1/3 deficiency, resulting in abnormal proinsulin processing [16]. Hypoglycemia is a possible consequence of glucagon deficiency; however, the pathophysiological mechanism of postnatal hyperinsulinemia in our patient is poorly understood. According to Zheng et al., hyperinsulinemia induces *PAX6* expression in endometrial epithelial cells, indicating that *PAX6* under insulin-resistant conditions may act as a potent transcriptional regulator and deregulate pathogenic mechanisms in patients with polycystic ovarian syndrome [16]. Since the HH episode was repeated but not persistent and no genetic mutation regarding congenital HH was identified, increased insulin secretion could have been caused by *PAX6* mutation-induced islet cell dysfunction, which subsequently led to HH during early childhood.

While the majority of published cases regarding *PAX6* mutations have focused on the clinical consequences of oculo- and neurogenesis, reports on its endocrinologic associations are scarce. Transitions to early onset DM have rarely been reported, and impaired glucose intolerance is mildly presented [17]. Motoda et al. reported a 63-year-old man with aniridia and insulin-dependent DM whose HbA1c was 9.7% and c-peptide was undetectable (c,483_486dupTTGG) [18]. Nishi et al. reported a similar case in a 27-year-old female (c.402del2) [17]. Both adult patients showed low insulin secretory capacity and ended up requiring lifelong insulin therapy. Our patient experienced a prediabetic condition (impaired glucose intolerance) at 60 months of age, and his age was the youngest ever reported. Hyperglycemia is a possible consequence of long-term diazoxide therapy, however the risk of developing early onset diabetes cannot be excluded even after the discontinuation of diazoxide therapy. Shimo et al. reported a 40-yearold woman with hypogonadotropic hypogonadism and borderline growth hormone deficiency and raised the possibility that a heterozygous *PAX6* mutation could induce partial hypopituitarism [19]. Screening for anterior pituitary hormones in our patient was normal for the same sex and age group. Although our patient showed normal insulin secretory capacity, close laboratory observation is critical to detect potential early onset DM.

Diazoxide, the first line treatment of congenital HH, binds to the linker region of the SUR1 subunit of the KATP channels and suppresses β-cell insulin secretion by forcing the channels to remain open [20]. Early diazoxide administration was a solid judgment for our patient during HH events, even prior to genetic diagnosis. However, excessive diazoxide therapy may increase the risk of DM or hyperglycemia; thus, its use in patients who are prone to glucose intolerance requires consideration [21]. Recent studies have proposed diazoxide as the first drug of choice for HH with a starting dosage of 5 mg/kg/day, which could be increased up to 20 mg/kg/day [2,22]. Due to the possible adverse effects of diazoxide, such as fluid retention, hypertrichosis, and cardiac failure, optimal dosage should be the minimal dosage as long as euglycemia is achieved [22]. In addition, the effect of diazoxide on glucagon-producing alpha cells is unknown. Intermittent hyperglycemia was a fortunate warning for our patient to discontinue diazoxide and reduce the risk of DM. In *PAX6*-mutated patients, any treatment that could affect glycemic control requires special attention, and blood glucose should be carefully monitored.

Additional information would have helped to clarify the role of *PAX6* in our patient. Serum glucagon levels were not measured due to the unavailability of the assay; hence, the documentation of hypoglucagonemia was not feasible, and glucagon compensation against hypoglycemia could not be evaluated. An 18-F-Dopa PET scan would have been useful to further detect the hyperinsulinemic cause. While the targeted NGS panel offered a clear conclusion, further genetic testing, such as whole exome or genome sequencing, may provide additional insights into our patient’s condition.

## 4. Conclusions

In conclusion, this case report details a *PAX6*-mutated anophthlamic boy manifesting with transient hypo- and hyperglycemia. *PAX6* mutations may lead to either life-threatening hypoglycemia or early onset glucose intolerance and diabetes. Further investigation is warranted to clarify the role of *PAX6* in aberrant glucose metabolism and islet cell hormone production.

## Figures and Tables

**Figure 1 medicina-57-00582-f001:**
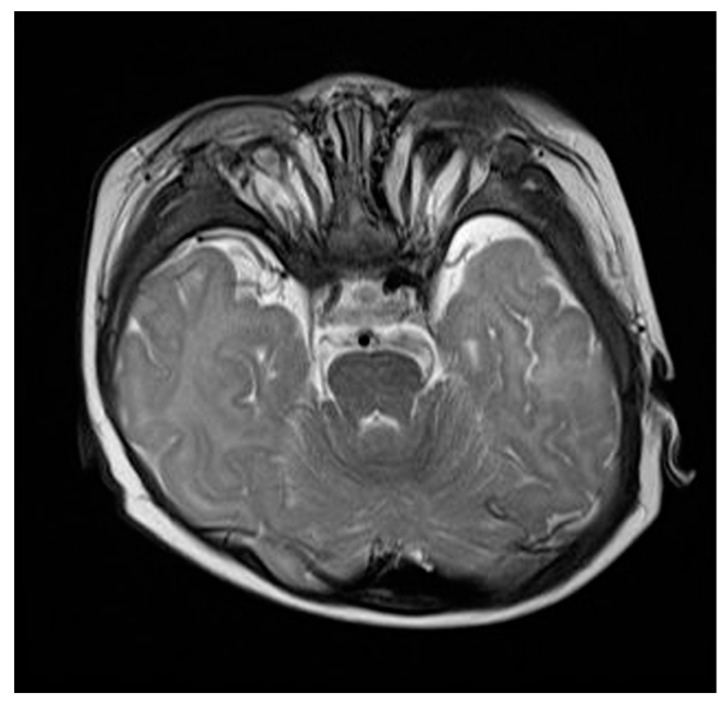
Brain magnetic resonance imaging at 6 months old demonstrated bilateral anophthalmia.

**Table 1 medicina-57-00582-t001:** Critical samples at two hypoglycemic seizure attacks at the age of 5 and 39 months, and fasting sample at the age of 60 months.

Parameters (Units) (Reference Range)	Age		
	5 Months	39 Months	60 Months
Glucose (mg/dL) (50.0–100.0)	25.0	38.0	156.0
Insulin (μU/mL) (hypoglycemic: <2.5; fasting: 1.0–30.0)	14.1	17.3	3.1
C-peptide (ng/mL) (1.1–3.3)	2.4	2.1	0.7
βOHB (μmol/L) (28.0–128.0)	24.1	27.2	30.8
HbA1c (%)(4.4–6.0)	4.0	4.6	6.1
FFA (μEq/L) (130.0–770.0)	193.0	181.6	161.0
Ammonia (μg/dL) (20.0–80.0)	72.0	80.6	51.0
Cortisol (μg/dL) (9.4–26.1)	8.6	12.1	8.8
pH (7.3–7.4)	7.4	7.4	7.3
HCO_3_^−^ (mmol/L) (23.0–30.0)	25.4	18.4	25.3

βOHB, beta-hydroxybutyrate; FFA, free fatty acids; HCO3^−^, bicarbonate; HbA1c, glycosylated hemoglobin.

## Data Availability

No new data were created or analyzed in this study. Data sharing is not applicable to this article.

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
