# Peer review of "Transient Hyperinsulinemic Hypoglycemia Linked to PAX6 Mutation"

_medicina, 2021, doi:10.3390/medicina57060582_

Round 1

Reviewer 1 Report

Are there any studies conducted with respect to PAX6 gene and glucose homeostasis? Please address those.

Author Response

We would like to thank the editor and reviewers for their comments on the manuscript. We tried to provide point-by-point response following the reviewer’s comments. Contents either added or rephrased were colored in red. We hope the editor and reviewers would agree on the revised version for the publication with improved quality.

Reviewer 2 Report

Here are my comments:

  1. The authors described the gene alteration detected as mutation. Why is it characterized as a mutation and not a single nucleotide polymorphism? Due to the frequency in the general population? Please review/ discuss in a bit more detail the PAX6 mutations/ polymorphisms previously described and associated with alterations in glucose metabolism and preferably separate them into those which are implicated in hyperinsulinemic hypoglycemia, diabetes or both.
  2. In order the manuscript to be able to be followed by a reader that is not familiar with genetic terminology, the authors are suggested to describe in words what the exact mutation means (for example substitution of a nucleotide etc) and also the result of the mutation at the level of protein (c.188C>G, p.Ser63Cys).
  3. What about the clinical examination of the liver and the penis in the patient? Any signs of skin pigmentation? Any history of jaundice? Were there any other abnormalities or syndromic features (for example those of WAGR syndrome which is attributed to mutations in the 11p chromosomal region (where PAX6 is also located)- except aniridia, kidneys or genitourinary anomalies/ tumors (Wilms tumor, gonadoblastoma), mental retardation, obesity etc. Did the authors correlate hypoglycemia with the temporal relation to meals?
  4. Did the authors measure GH, IGF-1, IGFBB1, TSH levels in the critical samples of hypoglycemia of their patient? What about lactate? And what about organic acids and amino-acids, or the presence of reducing substances in the urine? Have the authors measured the title of autoimmune antibodies for type 1 diabetes or performed an OGTT so far? These results would be interesting to show if exist.
  5. Please clarify that the glucagon was administered in the context of evaluation of hypoglycemia (as a dynamic test) and not additionally to the treatment of the first hypoglycemic attack (which was easily treated with intravenous glucose).
  6. Did the authors investigate their patient with a 18-F-Dopa PET scan?
  7. The authors administered a very low daily dose of diazoxide (5mg/kg/d). Please comment on that. Additionally, octreotide is another drug administered in these patients so, is it true that only diazoxide is FDA approved? Please clarify.
  8. What made the authors to think of discontinuing diazoxide at the age of 38 months? Did they have any clue that hyperinsulinism in their patient was temporary and not permanent or was it a trial?
  9. The authors had better make it clear that the possibility of developing diabetes in their patient in the future cannot be excluded even after the discontinuation of diazoxide treatment.
  10. Glucose levels are mg/dl. Please correct it in line 65.
  11. The authors are suggested to omit any general phrases from their abstract and decrease its length.
  12. There are some minor grammatical/ syntax errors throughout the text. A native English speaker could be helpful.

Author Response

We would like to thank the editor and reviewers for their insightful and constructive comments on the manuscript. We tried to provide point-by-point response following the reviewer’s comments. Former texts were described first, then contents either added or rephrased were colored in red. Consequently, reviewer’s comments have made the manuscript become more organized and logical. We hope the editor and reviewers would agree on the revised version for the publication with improved quality.
